# Emerging Technologies in Mass Spectrometry-Based DNA Adductomics

**DOI:** 10.3390/ht8020013

**Published:** 2019-05-14

**Authors:** Jingshu Guo, Robert J. Turesky

**Affiliations:** Masonic Cancer Center and Department of Medicinal Chemistry, University of Minnesota, 2231 6th St. SE, Minneapolis, MI 55455, USA; guoj@umn.edu

**Keywords:** Carcinogens, DNA adducts, adductomics, mass spectrometry

## Abstract

The measurement of DNA adducts, the covalent modifications of DNA upon the exposure to the environmental and dietary genotoxicants and endogenously produced electrophiles, provides molecular evidence for DNA damage. With the recent improvements in the sensitivity and scanning speed of mass spectrometry (MS) instrumentation, particularly high-resolution MS, it is now feasible to screen for the totality of DNA damage in the human genome through DNA adductomics approaches. Several MS platforms have been used in DNA adductomic analysis, each of which has its strengths and limitations. The loss of 2′-deoxyribose from the modified nucleoside upon collision-induced dissociation is the main transition feature utilized in the screening of DNA adducts. Several advanced data-dependent and data-independent scanning techniques originated from proteomics and metabolomics have been tailored for DNA adductomics. The field of DNA adductomics is an emerging technology in human exposure assessment. As the analytical technology matures and bioinformatics tools become available for analysis of the MS data, DNA adductomics can advance our understanding about the role of chemical exposures in DNA damage and disease risk.

## 1. Introduction 

Humans are frequently exposed to a wide variety of environmental and dietary genotoxicants and endogenously produced electrophiles. These reactive species can damage DNA and form covalent modifications, known as adducts. DNA adducts formed at critical sites in tumor-related genes are believed to be the first step in chemical carcinogenesis [1]. Selected DNA adducts have served as biomarkers of exposure and interspecies extrapolation of toxicity data [2,3,4]. Many pro-carcinogens require metabolic activation to form reactive intermediates before they covalently bind to DNA. These bio-activation reactions are catalyzed by xenobiotic metabolism enzymes that are broadly classified as Phase I or Phase II enzymes [5,6]. 

The reactive metabolites of genotoxicants and endogenously produced electrophiles modify DNA structure by alkylation [7], arylamination [8], bis-electrophile cross-link formation [9], and by adduction with reactive intermediates of lipid peroxidation [10,11] or free radicals [12]. The types of DNA adducts formed are dependent on the structures of the reactive chemicals, the nature of the electrophiles, and also on the ability of the compounds to intercalate with DNA, which may direct adduct formation to specific nucleophilic sites of the DNA bases. Several examples of chemicals and their DNA adducts are shown in Figure 1.

Adduct formation occurs at multiple nucleophilic sites of DNA (Figure 2), including the C8 atoms of guanine and adenine, endocyclic and exocyclic N and O atoms of the nucleobases [13]. The oxidation of the C5-methyl group of cytosine is an important marker of epigenetics [14], and the oxidation of the C8-atom of guanine produces 7,8-dihydro-8-oxoguanine (8-oxo-Gua) and 2,6-diamino-4-hydroxy-5-formamidopyrimidine (Fapy-Gua), which are prominent oxidative stress markers and genotoxic lesions [15,16,17]. Free radicals oxidize the C1′ and C4′ atoms of 2′-deoxyribose (dR), the C5-methyl group of thymine, and the C5 and C6 atoms of the pyrimidine ring [18]. DNA adducts of B[*a*]P, a carcinogenic polycyclic aromatic hydrocarbon (PAH), and NNK, a tobacco-specific nitrosamine and lung carcinogen, also form adducts on the phosphate backbone [19,20,21]. 

Many analytical techniques have been established to measure DNA adducts over the past three decades. The immunodetection and ^32^P-postlabeling assays established in the 1980s were among the first techniques to reveal that extensive DNA damage occurs in the human genome [22,23,24,25,26]. Other prominent analytical techniques include liquid chromatography (LC)-absorption/fluorescence spectroscopy [27] and LC-electrochemical detection [15,28]. Accelerator mass spectrometer, employing carcinogens with radiolabeled isotopes, has been used to detect DNA adducts in humans at ultra-trace levels [29]. However, these techniques are mostly indirect or non-quantitative methods and do not provide information about the structure of the adduct [30]. Gas chromatography with electron impact ionization mass spectrometry (MS), and negative ion chemical ionization has been employed to measure DNA adducts, particularly oxidized DNA bases, where the MS fragmentation spectra corroborate adduct structures [31,32]. More recently, LC coupled to electrospray ionization (ESI) has emerged as a breakthrough technology [33] that is capable of measuring a wide array of DNA adducts [13,34,35]. LC-MS has become a dominant platform in DNA adduct analyses. Recent advancements in instrument sensitivity have allowed for the measurements of DNA adducts formed with environmental genotoxicants and endogenously produced electrophiles in human DNA at levels as low as one adduct per 10^10^ nucleotides (nts) [36].

The approach of DNA adductomics, the simultaneous screening of multiple DNA adducts with no prior knowledge of the adducts in the samples, was first reported in 1990. The scanning of the neutral loss of the dR moiety was used to characterize the phenyl glycidyl-ether-modified nucleosides employing fast atom bombardment hybrid electric and magnetic sectors MS in both positive and negative ionization modes [37]. The cleavage of the glycosidic bond between the dR and the nucleobase is a commonly observed feature when 2′-deoxyribonucleosides (2′-dN) and 2′-dN adducts are subjected to the collision-induced dissociation (CID) in tandem MS. This feature has been widely applied in DNA adduct analyses and has served as a common scanning strategy for targeted measurements and non-targeted screening of multiple DNA adducts. This review will focus on the scanning strategies currently employed in the DNA adductomic experiments performed on different LC-MS platforms. The analytical challenges confronting the application of DNA adductomics in human studies where DNA adduct formation occurs at trace levels and the creation of bioinformatics tools needed to advance DNA adductomics technology are highlighted.

## 2. Current Scanning Strategies in DNA Adductomics

### 2.1. Key Features in LC-tandem MS-based DNA Adductomics

The LC-MS-based DNA adductomics analysis roots from the targeted DNA adduct measurement, in which a single or several adducts are scanned continuously in MS (Figure 3A). Adducts are usually recovered from macromolecular DNA following enzymatic hydrolysis using a cocktail of nucleases, which release the 2′-dN or phosphate adducts, or labile nucleobase adducts. For certain types of DNA adducts, acid or base hydrolysis, or neutral thermal hydrolysis are employed to release the nucleobase adducts from the DNA backbone [35,38].

The screening of chemically modified 2′-dN is advantageous for LC-MS-based DNA adductomics analysis because of the near-universal neutral loss of the dR (116 Da or 116.0473 Da in high-resolution accurate mass spectrometry (HRAMS)) upon CID. The cleavage of the glycosidic linkage of the DNA adduct precursor ion ([M + H]^+^) upon CID, produces the aglycone ion [M + H – 116]^+^ (or [B + H_2_]^+^) as the major fragment ion (Figure 4). This MS^2^ scan transition is a major feature for DNA adductomics scanning strategies. Other less common transition features include the neutral loss of the nucleobases or the detection of the nucleobase ions of the adducts, which have been employed to screen for DNA adducts derived from chemotherapeutic drugs or oxidatively damaged DNA (Section 2.3 and Section 2.7) [39,40]. 

### 2.2. Nomenclature 

#### Trap-Type CID and Beam-Type CID 

The mass spectrometers commonly used for DNA adduct analysis employ CID for tandem MS. The ions, typically protonated, are accelerated under an electrical potential in the gas phase, which results in increased kinetic energy and the collision with neutral gas molecules to generate fragment ions. Two types of CID are currently utilized in the contemporary mass spectrometers: (1) The trap-type CID found in ion trap and ion trap-hybrid instruments, and (2) the beam-type CID (also referred to as higher-energy CID (HCD) in hybrid Orbitrap instruments) employed in the collision cells of triple quadrupole (QqQ) and quadrupole (Q)-hybrid instruments, which include Q-time-of-fight (TOF), Q-trap, and Q-Orbitrap-hybrid instruments. Due to the design of the ion trap instruments (Section 2.4), ions are stored in the ion trap over an extended period, during which the ions are activated by applying pulsed electric fields at the resonant frequency. Trap-type CID via resonance excitation thus occurs as a function of time such that MS^n^ (with n ≥ 3) spectra can be acquired [41]. In beam-type CID instruments, ion packages pass the collision cells along the longitudinal axis of the device, and HCD-MS^2^ spectra are generated. The mass spectra produced by the two types of CID are similar in some respects, but are often complementary. For example, many DNA adducts lose the dR moiety when subjected to trap-type CID leaving the modified nucleobase (aglycone, [B + H_2_]^+^) as the sole product ion, regardless of how much resonance excitation is applied. An MS^3^ product ion scan is thus required for structure elucidation of the modified nucleobase portion of the adduct. In contrast, beam-type CID of DNA adducts can lead to more extensive fragmentation resulting in the formation of the aglycone ion and additional product ions, providing rich structural features of the modified base portion of the adduct at the MS^2^ scan stage [34]. In addition, HCD-MS^n^ spectra can be acquired on the Q-ion trap-hybrid-Orbitrap instruments, when an ion trap is used for ion selection.

#### Data-Dependent and Data-Independent Acquisition

Data-dependent (or information-dependent) acquisition (DDA or IDA) is a data collection mode widely used in proteomics and metabolomics analyses on Q-TOF, Q-trap, and hybrid Orbitrap MS. In DDA, a full MS scan is conducted over a *m/z* range, and certain ions, based on some predetermined rules (sometimes referred to as “filters”), are selected for the subsequent tandem MS (Figure 3D). Ion intensity range is a common filter applied in DDA, which is used alone or in combination with other filters, such as charge state selection and monoisotopic peak determination. In the “omics” analysis, a large number of ions may be eligible for the subsequent tandem MS. However, it is not feasible to perform fragmentation and collection of the resultant tandem MS spectra on all ions. One scanning strategy is to rank all ions based on their intensities from high to low and select only a fixed number of ions (“top N”) or select as many ions as possible within a defined duty cycle to perform the subsequent tandem MS. A dynamic exclusion function is often employed by restricting the reoccurrence of fragmentation events of the same precursor ions and thus, allowing more ions to undergo tandem MS. After a defined number of fragmentations (usually 1 to 3), the precursor ions are placed on an exclusion list for a designated time (exclusion duration) before the same precursor ions can be selected for fragmentation again. Both DDA and DDA combining constant neutral loss triggering MS^3^ (DDA-CNL-MS^3^, Figure 3E) have been utilized to scan for multiple DNA adducts (Section 2.4, Section 2.6 and Section 2.7). The performance of DDA is strongly influenced by the efficiency of ion transmission and scanning speed of the MS. DDA is biased towards the screening of abundant ions, and some DNA adducts present at low levels in the complex matrix may escape detection [42].

In contrast, data-independent acquisition (DIA) is an unbiased scanning technique that collects all precursor ions and their MS^2^ spectra within an *m/z* range. The identification of the detected analyte relies on the co-elution of precursor and fragment ions in the extracted ion current (EIC) chromatograms [42]. DIA typically contains one MS survey scan followed by one MS^2^ scan event covering the entire *m/z* range of interest (Figure 3F). More recently, Sequential Window Acquisition of all Theoretical Mass Spectra (SWATH) has gained popularity in the characterization and quantification of modified peptides and metabolites. SWATH fractionates the MS^2^ scan into evenly divided or variable mass windows to deconvolute (simplify) and improve the quality of MS^2^ spectra in the complex matrix (Figure 3G) [43]. We have modified SWATH-DIA and employed sectioned MS survey scans and corresponding MS^2^ scans to screen for DNA adducts of environmental genotoxicants and endogenously produced electrophiles (Figure 3H). DIA does not rely on the ion abundance, and data-mining can be performed retrospectively to probe for the precursor and aglycone ions of any potential DNA adduct without re-collecting data. 

#### Targeted and Untargeted Approaches

Both DDA and DIA can be operated in an “untargeted” fashion to scan for all possible adducts with no prior knowledge about the analytes present in the sample. However, targeted analysis, by the use of an inclusion list tabulated from the understanding of chemical exposures, the chemistry of DNA adduct formation, or reports from the literature can be utilized to guide adduct searching, thus improving the selectivity and sensitivity of detection. Similarly, an exclusion list can also be used to minimize the collection of undesired tandem MS of the background ions.

### 2.3. QqQ-MS

QqQ-MS is composed of three aligned quadrupoles in which the first (Q1) and the third quadrupoles (Q3) perform ion selection, and the middle quadrupole (q) serves as a collision cell to achieve beam-type CID fragmentation. The QqQ-MS has four basic tandem scanning modes: selected reaction monitoring (SRM), product ion scan, precursor ion scan, and constant neutral loss (CNL) scan [44]. The QqQ-MS in the SRM mode is used for targeted measurements of small molecules, including DNA adducts. The QqQ-MS is a widely used instrument because of its high sensitivity and selectivity, wide dynamic range, fast duty cycle, and robustness in operation [34]. The CNL, a modified SRM termed “pseudo-CNL”, and precursor ion scan have been employed as screening techniques in DNA adductomics. When utilizing CNL, Q1 and Q3 scan for adduct precursors and aglycones in a constant offset of 116 Da (loss of dR) over an *m/z* range encompassing the adduct precursor ions (Figure 3B and Figure 5). The CNL method can effectively detect adducts when the modification levels are relatively high, such as for synthetic standards [45]. DNA modified from treatment with carcinogens in vitro [46,47,48,49] and in vivo in animals dosed with high levels of carcinogens [50]. 

Matsuda’s laboratory employed the “pseudo-CNL” method to screen for multiple DNA adducts [51]. In pseudo-CNL, QqQ-MS detects the neutral loss of dR (116 Da) by monitoring a wide range of SRM transitions ([M + H]^+^ > [M + H -116]^+^) with single integer *m/z* increments covering the entire mass range of putative adducts (Figure 3C and Figure 5). As the QqQ-MS is a beam-type scanning instrument that detects ions of one *m/z* at a time, only 32 transitions were monitored in one sample injection to maximize the sensitivity of detection. Thus, each sample was injected 12 times to complete the total of 374 transitions. Nevertheless, there is a 10-fold or greater improvement in sensitivity when analyzing the same samples by pseudo-CNL compared to the CNL scan due to the slow duty cycle of the QqQ in full scan mode [42,52]. 

The mass spectral data are analyzed by the construction of a 2-dimension plot to visualize the DNA adductome map. The retention time is plotted on the *x*-axis, and *m/z* values are reported on the *y*-axis. The adducts are drawn as circles with the radius representing the normalized peak area against the internal standard of 2’-deoxyinosine. The DNA adductome map of human lung and esophagus tissues acquired by the pseudo-CNL scan method is illustrated in Figure 6. Many putative DNA adducts are detected, and some adducts appear to occur at different levels between the two tissues. However, it is not known whether these product ions are DNA adducts or simply other compounds that underwent the loss of 116 Da in tandem MS. The pseudo-CNL method was used to map DNA adducts in cells treated with genotoxicants including the plant constituent safrole [53,54]. soil bacteria exposed to acrolein [55], and genomic DNA extracted from the meat substitute Quorn, white buttom mushroom, and brewer’s yeast [56]. The pseudo-CNL also detected many putative DNA adducts of lipid peroxides in human organs [51,57,58,59]. However, these data should be interpreted with caution as the identities of these presumed adducts were not confirmed by full scan product ion spectra. Moreover, the artefactual formation of lipid peroxide DNA adducts may occur if antioxidants are not employed during DNA isolation. 

The precursor ion scan mode has also been employed to screen for DNA adducts. In this mode, a specific product ion is selected in Q3, and the precursor masses are scanned in Q1. All precursor ions that produce the same fragment ion detected in Q3 are thus identified. Inagaki and co-workers utilized the precursor ion scan to screen for nucleobase adducts with unknown chemical structures [39]. The protonated ions of guanine (Gua, *m/z* 152) and adenine (Ade, *m/z* 136) detected in Q3 were screened since these ions were major features in the product ion spectra of nucleobase adducts. The precursor ion scan mode employing Gua and Ade as target ions successfully detected aglycone adducts of glycidamide, a carcinogenic metabolite of acrylamide [60]. The CNL and precursor ion scan modes can be employed as a first screening approach to detect for putative DNA adducts in a sample. Thereafter, the identification of DNA adducts is achieved by acquiring fragmentation spectra collected in the product ion scan mode, and adduct quantification is accomplished in SRM using the stable isotope dilution method. Such an approach was utilized to detect 2′-deoxyguanosine (dG) adduct of 2-amino-1-methyl-6-phenylimidazo [4,5-*b*]pyridine (PhIP), a possible human carcinogen formed in well-done cooked meat [46,47,50], radiation-induced lesions in the DNA of γ-irradiated cells [48], adducts of PAHs [49], and the HAA, 2-amino-3-methylimidazo[4,5-*f*]quinoxaline (MeIQx) in rodent studies [61,62]. 

### 2.4. Ion Trap (IT)-MS

An IT is a device that “traps” ion through an oscillating electric field [44]. Depending on the configuration, an IT can store ions in 3-dimension, known as quadrupole ion trap (QIT), or in 2-dimension, which is referred to as a linear ion trap (LIT). The LIT has higher storage capacity and trapping efficiency than the QIT [63]. In contrast to QqQ-MS, where ion beams are transmitted continuously, an IT can store, select/isolate, fragment the ions of interest, and loop the whole operation sequentially in time to perform multistage (MS^n^) scanning (n ≥ 3) (Figure 7). However, IT has two drawbacks: a low mass cut-off and the potential of a space charge effect when too many ions enter the trap. The low mass cut-off, or so-called “1/3 Rule”, is an inherent phenomenon in which the IT fails to trap the ions at the lower end of the *m/z* range (below ~1/3 the *m/z* value of the precursor ion) during resonance excitation CID tandem MS [64]. This phenomenon can be partially mitigated by lowering the *q_z_* value (a parameter in IT that determines the ion trajectory stability inside the trap) to allow the trapping of low *m/z* ions; or by combining the IT with other mass analyzer to perform tandem MS, such as the q in Q-trap or the HCD cell in Q-LIT-hybrid Orbitrap MS (vide infra) [65,66]. The space charge effect occurs as the ion density increases in the trap resulting in the repulsion between ions of like charge, which causes the dispersion of the ion package and results in the drop-off of mass accuracy and sensitivity [44]. The influx of ions is tightly controlled in IT and IT-hybrid MS instruments by specifying the maximum allowed number of total ions that enter the traps utilizing automatic gain control (AGC) or by employing dynamic ion injection time, to minimize space charge effects. Increasing the AGC, within the allowable range, extends the intrascan dynamic range (the maximum abundance ratio between the most abundant and the least abundant signal observable within a given spectrum of ions collected in the trap) [67], thus increasing the likelihood to detect DNA adducts that are present at low abundance. The optimization of the AGC and injection times is necessary when operating an IT to maximize the performance of the instrument. 

IT has been used in many bioanalytical applications, including the characterization of drug metabolites [68], oligonucleotide sequencing [69], and identification of protein modifications [70]. DNA adducts formed with a wide variety of environmental carcinogens and endogenously produced electrophiles also have been identified and quantified by IT-MS [35,50,71]. Turesky was the first to utilize the DDA-CNL-MS^3^ scan in LIT to screen for multiple DNA adducts of different classes of genotoxicants (Figure 3E) [72]. The CNL mode, by the traditional definition, cannot be performed on the IT-MS. However, CNL-MS^3^ can be achieved through the DDA approach, whereby neutral loss of ion maps can be obtained retrospectively from the previous MS and MS^2^ scan data. The loss of dR (116 ± 0.5 Da) between adduct precursor ion (in MS) and aglycone ion (in DDA-MS^2^) triggered the subsequent MS^3^ scan if the aglycone was among the top 10 most abundant ions at the MS^2^ scan stage (Figure 8). Adducts were identified in human hepatocytes treated with the tobacco carcinogen 4-aminobiphenyl (4-ABP), and in the liver of rats treated with cooked-meat carcinogen MeIQx. DDA-CNL-MS^3^ with the LIT was applied to detect DNA adducts of the anticancer drug acylfulvene and illudin S induced DNA adducts in treated colon cancer cell lines [73]. The DDA-CNL-MS^3^ scanning is improved by usage of the HRAMS LIT-Orbitrap-hybrid MS, which is more selective in screening for DNA adducts by monitoring the loss of 116.0473 Da rather than the nominal mass at 116 Da (Section 2.7).

### 2.5. Q-Trap

The Q-trap MS is a hybrid QqQ-MS that has all standard scanning functions of a QqQ-MS and additional scanning features brought forth by the LIT replacing the Q3. In a Q-trap MS, the LIT can serve as a standard Q, or used as an additional fragmentation device to generate MS^3^ spectra, or to store the fragment ions from the collision cell (q) to improve the quality of product ion spectrum and the mass accuracy of the measured fragments (the enhanced product ion scan (EPI)) [66]. For applications in DNA adductomics, the Q-trap MS is first used in CNL mode for adduct discovery. The identity of the detected ion then can be obtained in a data-dependent mode with EPI (DDA-EPI) in the subsequent MS^2^ scan. The Feng laboratory employed this approach to detect DNA adducts of phenyl glycidyl ether, and styrene-7,8-oxide treated oligonucleotides [74]. They advanced this application by screening DNA of ovarian follicles treated with B[*a*]P or cigarette smoke condensate. Two B[*a*]P adducts were detected as results of two metabolic activation pathways of B[*a*]P, and one phenanthrene adduct was found as a degradation product of B[*a*]P [75]. Similarly, Chao and Cooke combined CNL with DDA-EPI to map adducts from DNA treated with the methylating agent methyl methanesulfonate (MMS) and its deuterated counterpart (d_3_-MMS). Five expected alkylated DNA adducts and three unknown adducts with 3 Da mass differences, and two putative endogenous adducts with the same *m/z* value in both MMS and d_3_-MMS treated samples, possibly derived from oxidative stress, were detected. They also screened for alkylated DNA adducts in liver DNA of mice individually dosed with five different *N*-nitrosamines. Five adducts were detected using the precursor ion scan mode, and their EPI spectra confirmed their identities based on published spectral data [76,77,78,79,80,81]. 

### 2.6. Q-TOF-MS

Q-TOF-MS can be viewed as a QqQ-MS with Q3 replaced by a high-resolution TOF mass analyzer. TOF separates ions according to their velocities acquired from an initial acceleration in an electric field when they drift through a field-free region to reach the detector [44]. In Q-TOF- MS, ions are measured with ppm-accuracy, which significantly improves the identification of analytes compared to the nominal resolution QqQ-MS, Q-trap-MS, and IT-MS. 

Van den Driessche employed Q-TOF-MS for the simultaneous detection of multiple DNA adducts formed with the chemotherapeutic drug melphalan in calf thymus DNA (CT DNA) treated in vitro and in a human lymphocyte cell line [82]. They employed DDA using the MS survey scan and MS^2^ with multiple collision energies to screen for compounds that had lost the dR moiety (116 Da) during CID. Adduct structures were corroborated from the consensus product ion spectra acquired at four collision energies. Thereafter, a ramped CID collision energy from low to high in the DIA mode (MS^E^) was applied to screen for DNA adducts in the lungs of mice exposed to nanosized-magnetite (MGT) [83]. A high-mass accuracy database containing over 100 adducts was constructed to facilitate the detection of potential DNA adducts. The adductome map revealed many putative adducts detected in lungs of mice treated with MGT compared to control mice. One proposed lipid peroxidation adduct, 3,*N*^4^-etheno-2′-deoxycytidine (εdC) was significantly elevated in MGT treated mice. The co-elution of the synthetic isotopically labeled internal standard was used as the criteria to confirm the identity of the adduct in the SRM mode using a QqQ-MS.

### 2.7. Quadrupole and LIT-Hybrid Orbitrap MS

The Orbitrap is an electrostatic ion trap that uses the fast Fourier transform to acquire mass spectra [84]. Ions are accumulated in an ion trap-like device (such as the “curved trap”, termed C-trap in the commercial hybrid Orbitrap from Thermo Fisher Scientific) for a predetermined period (max injection time) or until the influx ion count reaches the number defined by the AGC function [85]. Once injected into the Orbitrap; ions oscillate along the center electrode of the Orbitrap in a frequency measured by the acquisition of time-domain image current transients (transient length) before the conversion into a mass spectrum via Fourier transform [86]. Thus, longer transient length gives higher resolution, but with reduced scan speed. Since the Orbitrap is a trapping device, it is still vulnerable to the space charge effects. 

Currently, two classes of hybrid Orbitrap MS are available: The Q-hybrid MS and the Q-LIT-hybrid MS. The Q-hybrid MS has a front-end quadrupole performing ion selection and an HCD cell for beam-type CID MS^2^ fragmentation (HCD). The Q-LIT-hybrid MS has an additional dual-pressure LIT component that can perform MS^n^ fragmentation at nominal mass resolution. The Q-LIT-Orbitrap MS can easily switch between nominal mass resolution and high-resolution accurate mass detection (up to 1,000,000 resolution at *m/z* 200) and between HCD and trap-type CID. Newer versions of the hybrid Orbitrap also have the options for electron-transfer dissociation and ultraviolet photodissociation as additional fragmentation mechanisms. 

Hemeryck and colleagues screened for DNA adducts of dietary genotoxicants using high-resolution accurate mass measurement and ^12^C/^13^C ratio of the parent ion in full MS scan as detection criteria on a Q-hybrid Orbitrap. They employed an inclusion list of 123 DNA adducts to facilitate the adduct searching. The method was tested on CT DNA treated with alkylating agents, then extended to simulated in vitro gastrointestinal digestions containing cooked beef or poultry inoculated with fecal flora. In a follow-up study, genomic DNA was isolated from animals fed with cooked meat or poultry diet. The results demonstrated that the consumption of cooked meat altered the DNA adductome. The identities of the DNA adducts were propsed based on the high-resolution full MS data linking the precursor ions to DNA adducts postulated to form with genotoxicants present in red meat. Four adducts, *O*^6^-methylguanine (*O*^6^-MeG), *O*^6^-carboxymethylguanine (*O*^6^-CMG), pyrimidopurinone (M1G), and methylhydroxypropanoguanine (CroG) were tentatively identified on the basis of synthetic reference standards. However, stable isotopically labeled internal standards were not used in the study, and product ion spectra were not acquired to corroborate the structures of the proposed DNA adducts [87,88,89,90].

Balbo and co-workers extended the DDA-CNL-MS^3^ scanning method from the LIT-MS to HRAMS using a Q-LIT-hybrid Orbitrap MS [91]. The usage of an ultrahigh-performance LC (UHPLC) delivering sub-μL/min flow rates and a nanoESI source dramatically improved the sensitivity of the method. The method was applied to a mixture of 18 structurally diverse DNA adduct standards spiked in human leukocyte DNA, where the loss of dR (116.0473 Da ± 5 ppm) triggered the MS^3^ scan event. The method was then applied to liver DNA of mice exposed to NNK, in which multiple pyridyloxobutyl (POB)-DNA adducts and methylated DNA adducts were detected with MS^3^ spectra collected for structure confirmation. The on-line HPLC prefractionation of DNA adducts combined with the use of dynamic exclusion and an exclusion mass list containing non-modified nucleosides and their dimers dramatically improved the detection of DNA adducts present at low levels in the complex DNA digest matrix. This method was adapted to screen for adducts of the chemotherapeutic drug PR104A in treated colon epithelial cell lines. In addition to the CNL of dR, the neutral losses of the four bases (Gua, 151.0494 Da; Ade, 135.0545 Da; thymine, 126.0429 Da; and cytosine, 111.0433 Da) at 5 ppm mass tolerance were monitored in a targeted and untargeted manner to screen for expected and unexpected adducts [40,92]. 

The Turesky laboratory recently migrated the SWATH-DIA scanning strategy from proteomics and metabolomics to develop the wide-selected ion monitoring (wide-SIM)/MS^2^ to screen for DNA adducts using a Q-LIT-hybrid Orbitrap MS [42]. In order to improve the sensitivity of detection, the *m/z* range of potential DNA adducts, 330–630, was equally split into 10 sections. Each section comprised a wide-SIM scan of 30 *m/z* followed by an HCD-MS^2^ to scan for fragment ions (in *m/z* range 100 to 650) from all the precursor ions detected in the previous wide-SIM scan (Figure 3H). Both precursor ions in wide-SIM and MS^2^ product ions were detected with the high-resolution Orbitrap MS detector to achieve accurate formula assignment. Potential DNA adducts were characterized by co-elution of the extracted precursor ions ([M + H]^+^) in the wide-SIM scan and the aglycones ([M + H - 116.0473]^+^) in the succeeding MS^2^ scan at a mass tolerance of 5 ppm. A second injection performing targeted-MS^3^ of the aglycones was conducted, and the product ion spectra were compared to published mass spectral data to provide support for adduct identity. Wide-SIM/MS^2^ successfully detected adduct standards spiked into CT DNA digest matrix at levels 4 adducts per 10^9^ to 8 adducts per 10^8^ nts [42]. Thereafter, wide-SIM/MS^2^ detected DNA adducts of cooked meat carcinogen PhIP in human prostate, and a dA adduct formed with *N*-hydroxy-aristolactam-I (HONH-AL-I), a reactive metabolite of aristolochic acid-I (AA-I) in human kidney. AA-I is a urothelial carcinogen naturally occuring in some traditional Chinese herbal medicines [93]. The method also detected several DNA adducts of the tobacco carcinogen 4-ABP in human bladder (Figure 9) [42,94]. A constructed database with a list of 100 adducts served as a guide to identify for other potential adducts.

## 3. Challenges in DNA Adductomics in Human Samples

The goal of DNA adductomics is to assess the totality of DNA damage in the human genome, and by doing so, establish mechanistic linkages between chemical exposures and disease outcomes. The characterization of multiple adducts from DNA modified with high levels of carcinogens *in vitro* or the cell is readily accomplished by DNA adductomics approaches. However, in humans, the levels of DNA adducts formed with environmental genotoxicants or endogenously produced electrophiles are often at levels ranging from one adduct per 10^10^ to 10^8^ nts [35,36,95,96]. Thus, the screening of multiple DNA adducts and their identification poses great analytical challenges in human specimens, where the amounts of tissue are obtained in limited quantities. 

### Sample Availability

Fresh human tissue specimens are often not available for biomarker research because of the invasive nature of the biopsy procedure, and when accessible, only small amounts of tissue (low mg) and DNA (<10 µg) are obtained for assay. Moreover, the biopsy specimen may contain mixed cell types, and the level of DNA adducts may be diluted. Epithelial cells, for example, where more than 90% of human cancers originate [97], are the preferred cells for DNA adduct analyses. However, the composition of cells vary in biopsy samples containing epithelial or submucosal layers or in organs with glandular tissue containing a mixture of epithelium and stromal tissue (i.e., prostate and pancreas), especially in samples with hyperplasia. Several surrogate biospecimens have been used to screen for DNA adducts. These biospecimens include formalin-fixed paraffin embedded tissues [98], peripheral blood [99,100,101], buccal cells and saliva from the oral cavity [71,102,103], and epithelial cells exfoliated in urine [104,105] and breast milk [106].

The ESI signal responses vary among DNA adducts. Hence, some adducts are more amenable for analysis by DNA adductomics approaches. We have conducted spiking studies with CT DNA and successfully detected 20 DNA adducts by wide-SIM/MS^2^ at levels ranging from 0.4 to 8 adducts per 10^8^ nts, assaying 5 µg of DNA on column. Given this high level of sensitivity, it is feasible to conduct adductomics in biospecimens such as blood or saliva where more than 10 µg of DNA are recovered per mL of biological fluid. These measurements can provide information on exposures but may not be representative of the DNA damage that occurs in the target organ and the data must be interpreted with caution. 

### Sample Preparation and Data Collection 

Non-modified 2′-dN or nucleobases are present at one-million or higher levels than the DNA adducts of carcinogen. Thus, DNA adducts are often enriched by off-line solid-phase extraction (SPE), on-line HPLC fraction collection, liquid-liquid extraction, or on-line trapping to avoid issues that affect the LC-MS analysis, such as the overloading of the HPLC column, ion suppression, and space charge effects in the IT-MS [35,107,108]. Centrifugal filtration with a molecular weight cut-off filter can remove the enzymes used in DNA digestion [40], or to separate nucleobase adducts, such as depurinated *N*7-guanine adducts released from DNA backbone by mild hydrolysis [38]. High background ions can occur from the impurities in the chemicals and solvents, the materials in the frits of the SPE, and the membrane of centrifugal filters [50]. The use of recombinant enzymes can reduce some of the interfering ions introduced by enzymes of animal origin that may deteriorate the MS detection [42,91,108]. Extra precaution must be taken during sample preparation and storage when analyzing adducts formed by reactive oxygen species and lipid peroxidation, in which cases antioxidants should be added during the isolation of DNA to minimize artifact formation [109]. 

### Data Management

Hundreds of putative DNA adducts may be detected in the DNA adductomics analysis. The analysis and interpretation of DNA adductome data are challenging due to the sheer number of possible adducts. Comprehensive and reliable bioinformatic tools are required for the analysis of MS data of DNA adducts. An effective algorithm is in need for removal of background ions, the detection of precursor ions and their isotopic envelope for identification, formula assignment, fragment ion annotation, peak alignment between samples, and adduct list exportation. Universal procedures for automated data analyses are not currently available across different MS platforms for DNA adductomics. Some of the bioinformatics tools used in metabolomics can be migrated to DNA adductomics data analysis; however, a challenge remains when analyzing low levels of DNA adducts in complex matrices as usually occurs in humans. For example, the identification of a chemical in a metabolomics study is often based on the detection of an isotopic pattern or envelope containing the monoisotopic ([M_0_ + H]^+^), the first ([M_1_ + H]^+^) and the second isotopic peaks ([M_2_ + H]^+^). However, this approach can yield false negative results in DNA adductomics, particularly for adducts present in low abundance, where the corroborating isotopic peaks are below the limit of detection, or interfering ions preclude their detection (Figure 10). 

Several laboratories have established DNA adduct databases of the precursor ions and aglycones. We plan to establish an international consortium to develop a comprehensive, searchable database of exogenous and endogenous DNA adducts to facilitate DNA adduct data-mining and advance our understanding of disease risk and improve public health. This DNA adduct database will contain precursor ions, and fragmentation spectra library acquired at multiple collision energies on HRAMS platforms to facilitate data analysis and improve DNA adduct identification. 

## 4. Extension of DNA Adductomics Approach to the Emerging Fields of Urinary DNAa and RNA Adductomics

Urine is a non-invasive biospecimen, which has been employed to screen for modified nucleobases and 2′-dNs of exogenous exposures and endogenous biochemical processes [110,111,112]. Some of these urinary adducts are derived from intermittent chemical exposures and are transient, whereas other adducts resulting from frequent chemical exposures or oxidative stress may be continuously present at low steady-state levels. Adducts present in urine originate from oxidative DNA or RNA damage [113], labile alkylated adducts that undergo depurination, by DNA repair processes, or RNA turnover [114,115]. Cooke combined CNL and the precursor ion scan modes of the QqQ-MS to screen for 2′-dN and nucleobase adducts, respectively, in urine. Six nucleoside adducts and 10 nucleobase adducts were spiked into a urine sample from a healthy subject. A urinary adductome map similar to that from the pseudo-CNL scan was generated with integrated peak area normalized against urinary creatinine. The scanning strategy detected all 2′-dN adducts and 8 out of 10 nucleobase adducts. After that, an adductome map of urine from a patient under mechanical ventilation in an intensive care unit was compared to the adductome map of one healthy subject. More putative adducts were detected and at a higher intensity in the urine of a patient compared to the healthy individual [116].

RNA is also modified through genotoxicants and reactive endogenous electrophiles. There are few reports on the biomonitoring of RNA adducts; however, aflatoxin B_1_ (AFB_1_)- and AA-I-RNA modification levels are higher than those for DNA [117,118]. RNA modification is a continuously occurring phenomenon in all organisms. The modifications are dynamic, reversible, pivotal to cell functions, and some RNA adducts have been linked to human diseases [119,120,121,122]. Primary interests in RNA modification have focused on the identification of the sites of adduct formation and their impact on RNA transcription [123]. More than 160 RNA modifications have been identified with chemical structures, biosynthetic pathways, and sequence information [124]. The techniques employed in DNA adductomcs can also be applied to the screening of RNA adducts formed with genotoxicants and endogenous electrophiles. 

Mass spectrometry has played a crucial role in the identification of DNA adducts in humans, and the biomarker data strengthen the associations between chemical exposures and disease risk [125,126,127,128,129]. AFB_1_ and AA-I are two examples where chemical exposures, chemical-specific DNA adducts in target tissues, combined with mutation spectra in tumor-related genes have provided a mechanistic understanding of the causal roles for these chemicals in the development of liver and upper urothelial cancers, respectively [1,93,130]. DNA adductomics is a relatively new and developing technology. When the analytical methods are fully mature and robust bioinformatics tools are available for the analysis of MS data, DNA adductomics technologies can be implemented in molecular epidemiology studies to further our understanding of the linkage between chemical exposures, DNA damage, and disease outcomes. 

## Figures and Tables

**Figure 1 high-throughput-08-00013-f001:**
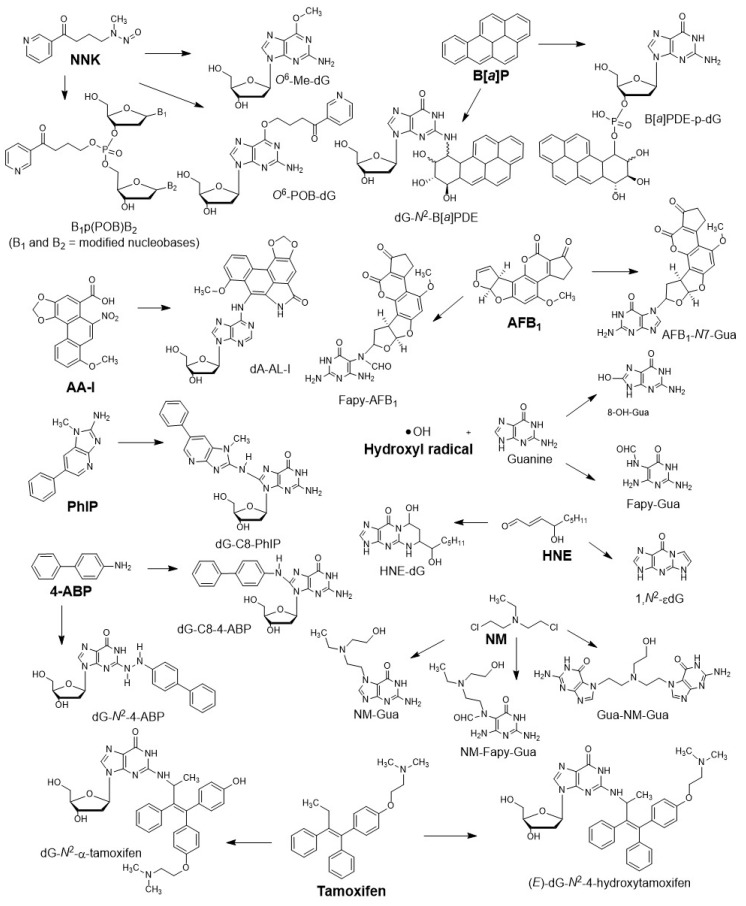
Structures, names, abbreviations, and DNA adducts of representative genotoxicants or carcinogens: 4-(methylnitrosamino)-1-(3-pyridyl)-1-butanone (NNK), a tobacco-specific nitrosamine; benzo[*a*]pyrene (B[*a*]P), a polycyclic aromatic hydrocarbon; aflatoxin B_1_ (AFB_1_), a mycotoxin; aristolochic acid -I (AA-I), a component in herbal medicine; 2-amino-1-methyl-6-phenylimidazo [4,5-*b*]pyridine (PhIP), a heterocyclic aromatic amine and possible human carcinogen; 4-aminobiphenyl (4-ABP), an aromatic amine; hydroxyl radical (•OH), an endogenous electrophile; (*E*)-4-hydroxynon-2-enal (HNE), an endogenous lipid peroxidation product; bis(2-chloroethyl)-ethylamine, a nitrogen mustard (NM), a chemotherapeutic drug; tamoxifen, a hormone therapy drug.

**Figure 2 high-throughput-08-00013-f002:**
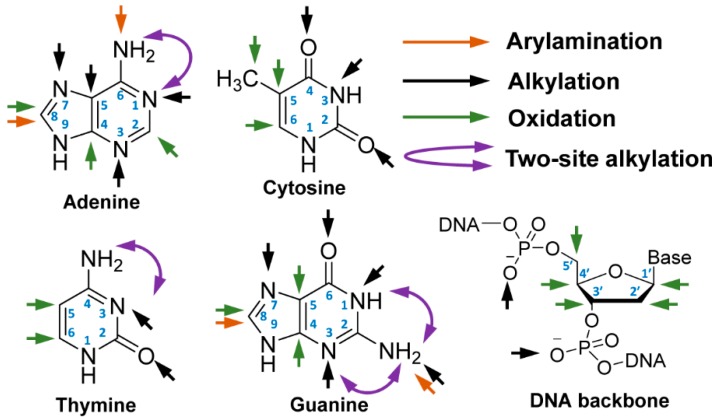
Reactive sites for adduction and oxidative damage of DNA. (Liu et al. Chem. Soc. Rev. 2015. Reproduced by permission of The Royal Society of Chemistry).

**Figure 3 high-throughput-08-00013-f003:**
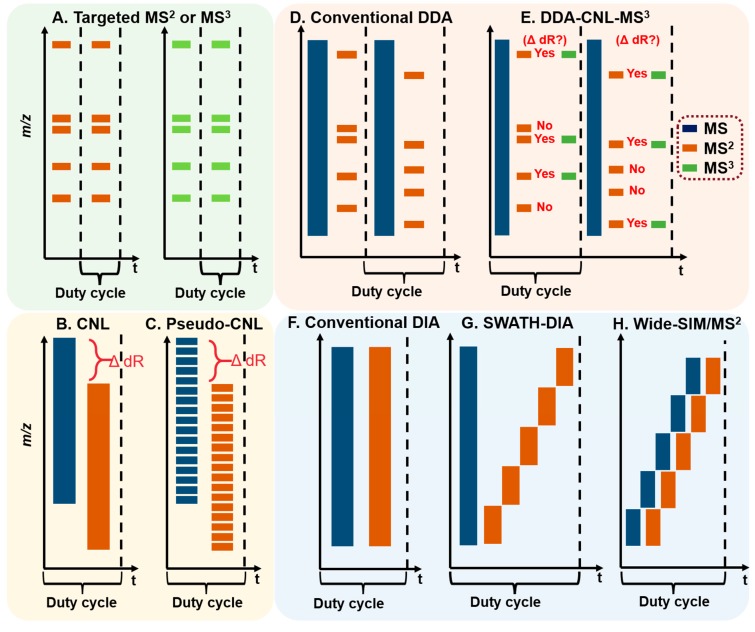
Scanning strategies employed in DNA adduct analyses. DNA adducts are quantified by (**A**) targeted MS^2^ or MS^3^ scans by the stable isotope dilution method using a QqQ or IT MS. On the QqQ MS platform, DNA adductomics can be performed in (**B**) CNL mode, in which Q1 and Q3 scan continuously with a mass offset of 116 Da (corresponding to the loss of dR), or (**C**) in a modified SRM mode, known as pseudo-CNL, in which multiple [M + H]^+^ > [M + H -116]^+^ transitions with integer increment are employed over the entire *m/z* range to scan for putative 2′-dN DNA adducts. The DDA and DIA approaches utilized in proteomics and metabolomics have been applied to DNA adductomics with modifications. In (**D**,**E**) DDA, precursor ions are detected in the MS survey scan, and a certain number of ions are selected for MS^2^ fragmentation based on predetermined conditions, such as ion intensity range or neutral loss. On the Q-LIT-MS or LIT-hybrid MS platforms, 2′-dR DNA adducts are detected and structures confirmed with (E) DDA-CNL-MS^3^, where the neutral loss of dR triggers the fragmentation of aglycone ions. In DIA approaches (**F–H**), all precursor ion undergo fragmentation (**F**) in one scan event, and the identification of a DNA adduct relies on the co-elution of the precursor and fragment ions. (**G**) SWATH-DIA divides the MS scan into smaller windows, thereby simplifying the quality of MS^2^ spectra in the complex matrix. The modified SWATH-DIA, (**H**) wide-SIM/MS^2^, has been used to scan for multiple DNA adducts, in which both MS and MS^2^ are separated into smaller windows, which improves the sensitivity of detection of precursor adducts and the aglycones.

**Figure 4 high-throughput-08-00013-f004:**
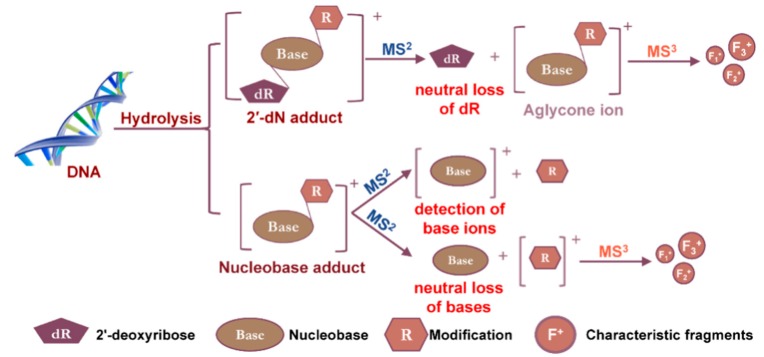
Fragmentation scheme of 2′-dN and nucleobase adducts in tandem MS. Features that have been used in DNA adductomic screening are highlighted in red.

**Figure 5 high-throughput-08-00013-f005:**
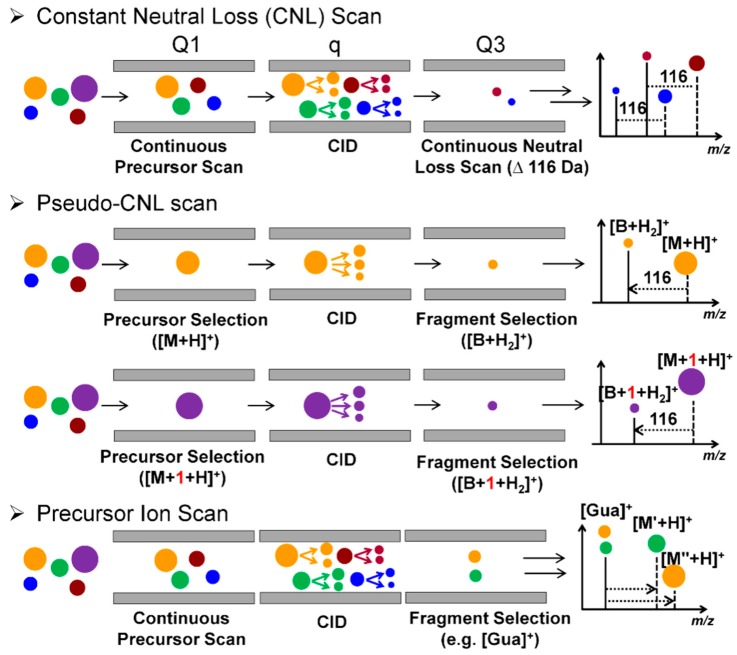
Scan modes used in DNA adductomics analysis by QqQ-MS. In the CNL mode, Q1 and Q3 scan continuously with a *m/z* offset of 116 Da that corresponds to the loss of dR. The pseudo-CNL scans [M + H]^+^ to [M + H -116]^+^ transitions with one Da increments to cover the entire *m/z* range for scanning putative adducts. The precursor ion scan detects DNA adduct precursors that generate common fragment ions, such as [Gua]^+^ and [Ade]^+^. Ions of different *m/z* are depicted by the size of circles.

**Figure 6 high-throughput-08-00013-f006:**
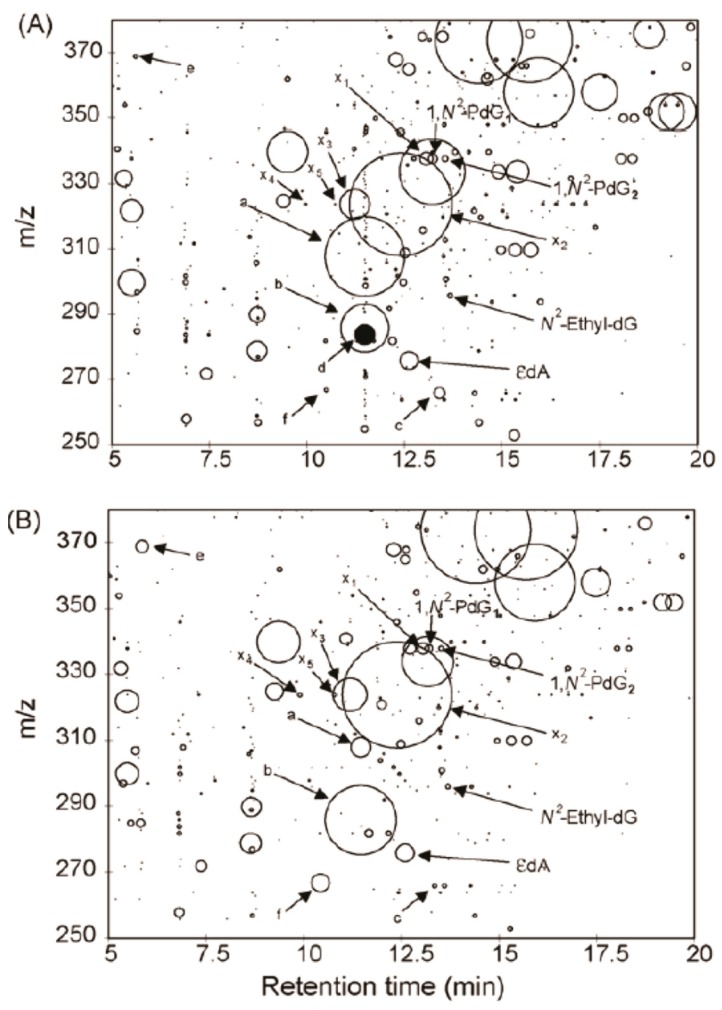
Adductome map of putative DNA adducts detected in human (**A**) lung and (**B**) esophagus tissues using the pseudo-CNL scan mode in a QqQ-MS. The adducts are drawn as circles with the radius representing the normalized peak area against the internal standard of 2’-deoxyinosine. Four adducts *N*^2^-ethyl-2′-deoxyguanosine (*N*^2^-ethyl-dG), α-S- and α-R-methyl-γ-hydroxy-1,*N*^2^-propano-2′-deoxyguanosine (1,*N*^2^-PdG1 and 1,*N*^2^-PdG2), and 1,*N*^6^-etheno-2′-deoxyadenosine (ɛdA) were identified by comparison to authentic standards and by the stable isotope dilution method in the SRM scan mode. Putative adducts are labeled x1 through x5 and a through f with an arrow. (Kanaly et al. Mutat Res, 2007. Reproduced by permission of Elsevier.).

**Figure 7 high-throughput-08-00013-f007:**
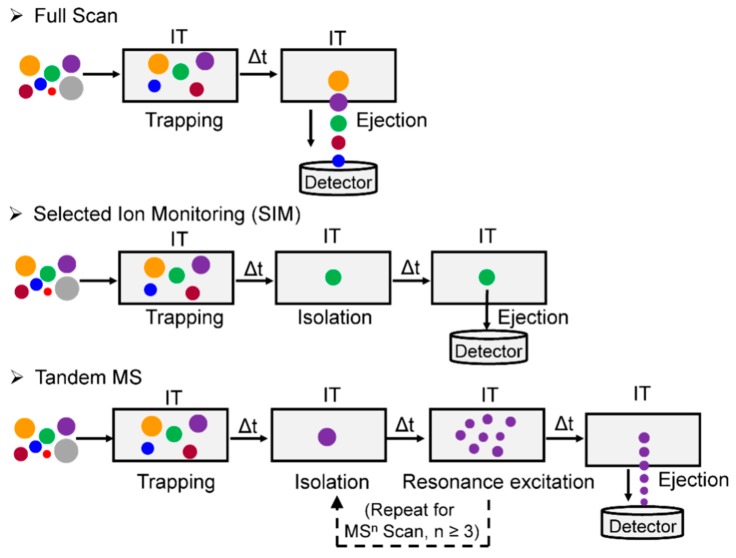
Scan modes of IT-MS. An IT can store and transmit ions over a large *m/z* range or isolate ions of a very narrow *m/z* (SIM, usually between 1 and 5 *m/z* window). The selected ions can be ejected for detection directly or undergo resonance excitation fragmentation. The isolation of ions and fragmentation can be repeated multiple times to acquire MS^n^ spectra. Ions of different *m/z* are depicted by the size of circles. The time-lapse among the different functions are indicated as Δt. (Guo et al. Curr. Protoc. Nucleic Acid Chem., 2016. Reproduced by permission of John Wiley and Sons.).

**Figure 8 high-throughput-08-00013-f008:**
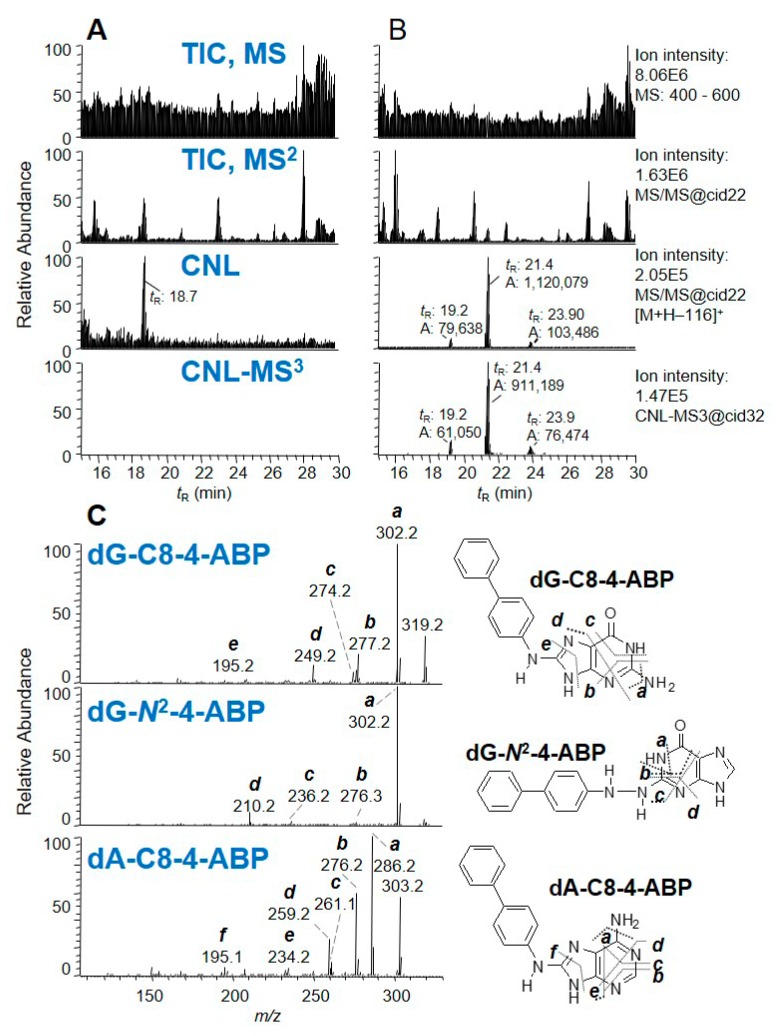
DDA-CNL-MS^3^ chromatograms of 4-ABP adducts in human hepatocytes (**A**) untreated and (**B**) treated with 4-ABP. The TIC chromatograms of MS and MS^2^ scans are shown on the top rows, followed by the EIC of MS^2^ scans with the CNL of 116 Da and EIC of CNL-triggered MS^3^ scans. The structures of the three detected 4-ABP adducts were corroborated from the (**C**) MS^3^ spectra in comparison to the published spectral data. (Bessette et al. Anal. Chem. 2009. Reproduced by permission of American Chemical Society.).

**Figure 9 high-throughput-08-00013-f009:**
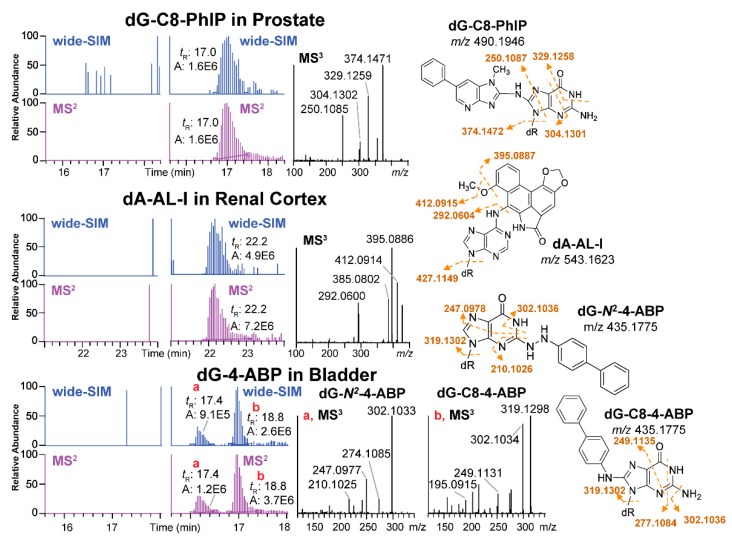
The detection of DNA adducts in human biopsy samples by wide-SIM/MS^2^: dG-C8-PhIP was detected in prostate; dA-AL-I was detected in kidney; and dG-C8-4-ABP and dG-*N*^2^-4-ABP, in bladder. Adduct structures (shown to the right) were confirmed by the HRAMS measurement with 5 ppm mass tolerance and MS^3^ spectra compared with the published data. (Guo et al. Anal Chem 2017, Guo et al. Chem Res Toxicol, 2018. Reproduced by permission of American Chemical Society.).

**Figure 10 high-throughput-08-00013-f010:**
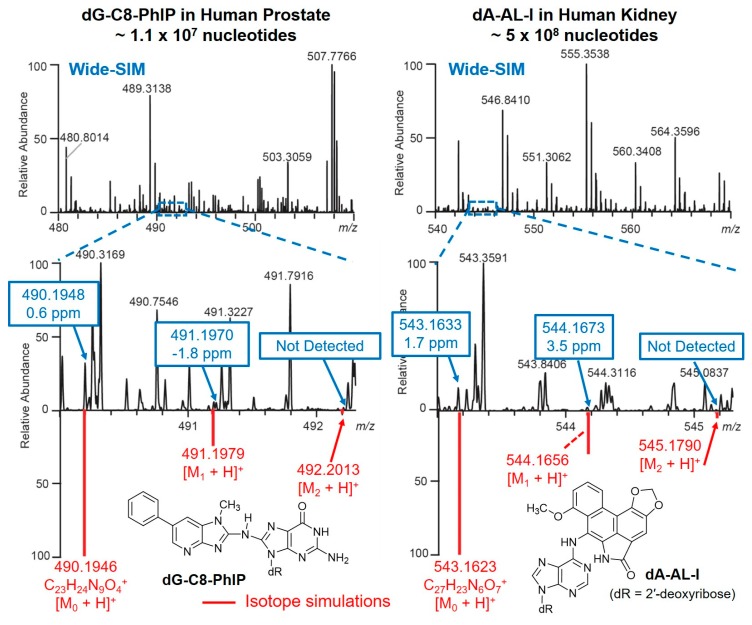
The precursor ions of dG-C8-PhIP and dA-AL-I were detected in human biopsy samples using the wide-SIM/MS^2^ method. The *m/z* values from the MS measurement and the *in silico* simulations of the monoisotopic ([M_0_ + H]^+^), first ([M_1_ + H]^+^), and second isotopic peaks ([M_2_ + H]^+^) of dG-C8-PhIP and dA-Al-I are highlighted in blue and red, respectively. The second isotopic peaks of both adducts were not detected due to the low ion abundance.

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
