# Peer review of "Emerging Technologies in Mass Spectrometry-Based DNA Adductomics"

_2571-5135, 2019, doi:10.3390/ht8020013_

Round 1
Reviewer 1 Report
DNA adductomics is a powerful tool to measure the DNA damages arisen from environmental and every day exposures. Knowing the important role mass spectrometry has played and with the improvement in mass spectrometry instrumentation such as the sensitivity and selectivity, we are now heading closer to the comprehensive measurement of DNA damages by identifying more putative DNA adducts in human tissues.
This manuscript is a sequel of a review published in 2014 with the same title. This manuscript is well written and covers the up-to-date knowledge and application of different MS scanning techniques that have been recently used in DNA adductomics with detailed description. The authors also addressed the challenges commonly faced in DNA adductomics in human samples and called for international cooperation to develop a comprehensive database for DNA adducts that is also inspiring.
Overall, this is an informative manuscript that is very useful for researchers in relevant fields.
Author Response
We appreciate the reviewer for the positive feedback and the support for the publication of our review.
Reviewer 2 Report
DNA adductomics is a small field and this is the third review in five years (the two previous reviews are by Balbo et al., Chem Res Toxicol. 2014 and Villalta et al., Int J Mol Sci. 2017). Much of this review covers the same concepts and topics, and it is bound to feel repetitive for those of us who have read the earlier reviews. However, the present review have many merits and since this is a special issue on adductomics I think it has a given place. Perhaps the title could be changed or a subtitle added since the review by Balbo et al. from 2014 share the same title? I think this review should have a unique title.
As a review it is a very informative and educational and for a person new to the topic I think it should be considered recommended reading. I have some minor comments regarding the figures; I am fine with the use of figures by other authors but many of the figures in the review suffer from low resolution (most importantly figures 2, 3,4,6, 9 and 10) and I think the review would benefit from clearer figures with higher resolution.
In this review I especially enjoyed the third section, "Challenges in DNA adductomics in human samples". It would be interesting if the authors could discuss whether DNA adductomics is a practical tool for biomarker discovery at the present state of methodologies and instrumentation? If one was to perform DNA adductomics using human blood samples, how much blood would be needed? I assume a volume of about 10 mL would be needed, which may not be feasible when working with samples from valuable cohorts when the sample volumes that may be provided for such analysis is often very limited. For me, this stands out as the major limiting factor for DNA adductomics. Do the authors think this limitation could be overcome and, if so, how? Also, it would be interesting if the authors mentioned the parallels with methods for protein adductomics and how such methods could complement each other.
Overall, I think the review is acceptable for publication following some minor changes.
Author Response
We appreciate the reviewer for the positive feedback and the support. Here are the responses to the comments.
The title of the review has been changed to "Emerging Technologies in Mass Spectrometry-based DNA Adductomics"
We have updated the figures with better resolution.
A paragraph in Section 3 has been added addressing the feasibility of conducting DNA adductomics in blood. "The ESI signal responses vary among DNA adducts. Hence, some adducts are more amenable for analysis by DNA adductomics approaches. We have conducted spiking studies with CT DNA and successfully detected 20 DNA adducts by wide-SIM/MS2 at levels ranging from 0.4 to 8 adducts per 108 nts, assaying 5 µg of DNA on column. Given this high level of sensitivity, it is feasible to conduct adductomics in biospecimens such as blood or saliva where more than 10 µg of DNA are recovered per mL of biological fluid. These measurements can provide information on exposures but may not be representative of the DNA damage that occurs in the target organ and the data must be interpreted with caution."
Reviewer 3 Report
Review of the manuscript
Guo J. and Turesky R.J.
DNA Adductomics
Submitted to High-throughput
Principal comments
The submitted manuscript is a review on the state-of-the-art of DNA adductomics written by leading researchers in the field. It is a very well organized, systematic and clearly written paper. The topics is presented in sufficient but not tedious details. The major focus is on the mass spectrometric scanning strategies, with examples of their use in various specific carcinogen-DNA adduct investigations. I found no weekness of the manuscript to be pointed out and recommend its publication in its current form, with only few minor comments (below) to be considered.
Detailed comments
1) Pg. 3, Fig. 2 It would be useful to show numbering of atoms in the nucleobase molecules (for better understanding of names of the particular adducts presented throughout the paper).
2) Pg.3, line 66: „…employing radiolabeled isotopes of carcinogens“ I recommend a more accurate wording „employing carcinogens with radiolabeled isotopes“.
3) Pg. 7, line 194: section 2.2 QqQ-MS should be numbered 2.3. The same note for all subsequent sections in chapter 2.
4) Pg. 6, line 175: „The identity of the detected analyte relies on…“ better: „The identification of the detected analyte relies…“
5) Pg. 9, Figure 6: Meaning of size of the circles in Fig. 6 should be described directly in the figure caption (in addition to pg. 8, line 227).
6) Pg. 11, Fig. 7: Constant neutral loss (CNL) is not included among the scanning modes used in IT-MS, but it is mentioned in line 300 that Turesky utilized DDA-CNL-MS3 scan in LIT. Please clarify whether CNL mode is applicable in IT or under which circumstances.
7) Pg. 14, text and Fig. 9: The abbreviation for aristolochic acid is AA but symbol AL is used in the Fig. 9. Please explain in the figure caption or close to it that AL stands for aristolactam.
8) Pg. 15, line 444: The sentence „However,…“ is missing a predicate.
9) Pg. 15, line 456: „Centrifugal filtration …can remove the enzymes used in DNA digestion or to recover nucleobase adducts after mild acid or base hydrolysis. It is not clear to me how centrifugal filtration can recover nucleobase adducts or what this recovering means.
14.4.2019
Author Response
We appreciate the support and the detailed comments from the reviewer. Here are our responses:
1) atom numbers have been added to Figure 2
2) sentence has been rewritten as suggested
3) section numbers from 2.3 to 2.7 in the text and the section titles have been corrected
4) corrected
5) We have added an extra line stating "The adducts are drawn as circles with the radius representing the normalized peak area against the internal standard of 2'-deoxyinosine" to the legend of Figure 6.
6)To clarify, this part were rewritten as "Turesky was the first to utilize the DDA-CNL-MS3 scan in LIT to screen for multiple DNA adducts of different classes of genotoxicants (Figure 3E).[72] The CNL mode, by the traditional definition, cannot be performed on the IT-MS. However, CNL-MS3 can be achieved through the DDA approach, whereby neutral loss of ion maps can be obtained retrospectively from the previous MS and MS2 scan data. The loss of dR (116 ± 0.5 Da) between adduct precursor ion (in MS) and aglycone ion (in DDA-MS2) triggered the subsequent MS3 scan if the aglycone was among the top 10 most abundant ions at the MS2 scan stage (Figure 8)."
7)The metabolism of aristolochic acid-I (AA-I) generates N-hydroxy-aristolactam-I (AL-I) that can form DNA adducts. Thus we have specified in the text on top of Figure 9 that "... and a dA adduct formed with N-hydroxy-aristolactam-I (HONH-AL-I), a reactive metabolite of aristolochic acid-I (AA-I) in human kidney. AA-I is a urothelial carcinogen naturally occuring in some traditional Chinese herbal medicines.[93]"
8) We have added the word "vary" in the referred sentence.
9) Here we are referring to two circumstances. First, DNA was digested with an enzyme cocktail to nucleosides and the centrifugal filtration was used to remove the enzymes (see reference 40). Second, some nucleobases and base adducts can be released from DNA after acid/base hydrolysis treatment. The centrifugal filtration was used to remove the DNA backbones, which was large enough to not pass through the mass cut-off membrane (see reference 38). Thus, we have rewritten the sentence to "Centrifugal filtration with a molecular weight cut-off filter can remove the enzymes used in DNA digestion,[40] or to separate nucleobase adducts, such as depurinated N7-guanine adducts released from DNA backbone by mild hydrolysis.[38]"